# The Archaeome’s Role in Colorectal Cancer: Unveiling the DPANN Group and Investigating Archaeal Functional Signatures

**DOI:** 10.3390/microorganisms11112742

**Published:** 2023-11-10

**Authors:** Nour El Houda Mathlouthi, Imen Belguith, Mariem Yengui, Hamadou Oumarou Hama, Jean-Christophe Lagier, Leila Ammar Keskes, Ghiles Grine, Radhouane Gdoura

**Affiliations:** 1Laboratoire de Recherche Toxicologie Microbiologie Environnementale et Santé (LR17ES06), Faculté des Sciences de Sfax, University of Sfax, Sfax 3000, Tunisia; nourhouda0810@gmail.com (N.E.H.M.); yenguiimariem1995@gmail.com (M.Y.); 2Laboratoire de Recherche de Génétique Moléculaire Humaine, Faculté de Médecine de Sfax, University of Sfax, Avenue Majida BOULILA, Sfax 3029, Tunisia; mayno87.ib@gmail.com (I.B.); ammarkeskesl@gmail.com (L.A.K.); 3IHU Méditerranée Infection, l’unité de Recherche Microbes, Evolution, Phylogénie et Infection (MEPHI), 19-21, Bd. Jean Moulin, 13005 Marseille, France; hamadouh2o@gmail.com (H.O.H.); jean-christophe.lagier@univ-amu.fr (J.-C.L.); grineghiles@gmail.com (G.G.); 4Institut de Recherche pour le Développement (IRD), Aix-Marseille Université, IHU Méditerranée Infection, l’unité de Recherche Microbes, Evolution, Phylogénie et Infection (MEPHI), 13005 Marseille, France

**Keywords:** colorectal cancer, gut microbiome, archaea, DPANN group, MvhB-type polyferredoxin, *Ferroglobus placidus*, *Natrinema* sp. *J7-2*, archaeal biomarker

## Abstract

Background and Aims: Gut microbial imbalances are linked to colorectal cancer (CRC), but archaea’s role remains underexplored. Here, using previously published metagenomic data from different populations including Austria, Germany, Italy, Japan, China, and India, we performed bioinformatic and statistical analysis to identify archaeal taxonomic and functional signatures related to CRC. Methods: We analyzed published fecal metagenomic data from 390 subjects, comparing the archaeomes of CRC and healthy individuals. We conducted a biostatistical analysis to investigate the relationship between *Candidatus Mancarchaeum acidiphilum* (DPANN superphylum) and other archaeal species associated with CRC. Using the Prokka tool, we annotated the data focusing on archaeal genes, subsequently linking them to CRC and mapping them against UniprotKB and GO databases for specific archaeal gene functions. Results: Our analysis identified enrichment of methanogenic archaea in healthy subjects, with an exception for *Methanobrevibacter smithii*, which correlated with CRC. Notably, CRC showed a strong association with archaeal species, particularly *Natrinema* sp. *J7-2*, *Ferroglobus placidus*, and *Candidatus Mancarchaeum acidiphilum.* Furthermore, the DPANN archaeon exhibited a significant correlation with other CRC-associated archaea (*p* < 0.001). Functionally, we found a marked association between MvhB-type polyferredoxin and colorectal cancer. We also highlighted the association of archaeal proteins involved in the biosynthesis of leucine and the galactose metabolism process with the healthy phenotype. Conclusions: The archaeomes of CRC patients show identifiable alterations, including a decline in methanogens and an increase in Halobacteria species. MvhB-type polyferredoxin, linked with CRC and species like *Candidatus Mancarchaeum acidiphilum*, *Natrinema* sp. *J7-2*, and *Ferroglobus placidus* emerge as potential archaeal biomarkers. Archaeal proteins may also offer gut protection, underscoring archaea’s role in CRC dynamics.

## 1. Introduction

Colorectal cancer (CRC) is considered to be one of the deadliest cancers worldwide, with an estimated incidence of 1.9 million cases in 2020 [1]. The gut microbiota appear to be one key factor in CRC tumorigenesis [2]. It is a complex ecosystem that includes many microorganisms hosted in the digestive tract, mainly bacteria, viruses, fungi, and archaea.

An impairment of the gut microbiota, referred to as dysbiosis, can favor the colonization of the intestinal mucosa by opportunistic microbes [3], which could be involved in carcinogenesis according to a “driver-passenger” model. The first stage consists of colon colonization by pathogenic “driver” bacteria with pro-inflammatory and pro-carcinogenic potential, such as *Bacteroides fragilis* and *Escherichia coli* species, contributing to the initiation of CRC [4]. The second step involves opportunistic “passenger” bacteria, such as *Fusobacterium nucleatum*, leading to tumor growth and its progression via interactions with the host immune system or the production of metabolites [5].

Viromes were also found to be changed when comparing the late and early stages of CRC. Advanced CRC microbiomes were enriched with several bacteriophages such as Betabaculus virus and Mulikevirus [6]. In addition, the abundance of human papillomavirus (HPV) and bocavirus was found to be significantly higher in tumoral tissues compared to adjacent tissues [7]. It was proposed that colon viruses could induce DNA alteration and alter immune homeostasis [6,8].

CRC is also linked to a fecal fungal dysbiosis presented by an abundance of Basidiomycetes and Ascomycetes as well as opportunistic fungi such as Malassezia. In contrast, the depletion of Saccharomycetes such as *Saccharomyces cerevisiae* was highlighted. This fungal species was found to exhibit regulatory and anti-inflammatory effects on the host by inducing interleukin-10 production [9,10].

However, due to the difficulty of culturing, detecting, and identifying it, archaea have remained “a forgotten player in the human microbiota” [11,12], and researchers only recently brought into focus the importance of the archaeome in CRC [13].

Archaea are prokaryotic unicellular microorganisms, and their entirety, including their genetic material in a particular environment, is called an archaeome [14]. Some of these microbes are characterized by membranes built with ether/isoprenoid chains of L-glycerol, and their cell walls lack peptidoglycan [15], while others are shared with eukaryotes or bacteria. The archaea were mainly considered extremophiles, residing in environments foreign to the human habitat and the microbial ecosystem. Thanks to the progress of molecular methods, especially the development of DNA sequencing and metagenomics, the presence of archaea in non-extreme environments has been revealed [16].

Indeed, metagenomic analyses highlighted its presence in the human microbiome and its ability to behave as a pathogen. Among its characteristics that may mediate human pathogenicity, we cite the ability to access and colonize the human microbiome and to compete with other endogenous microbes for survival, inducing dysbiosis [16], and the possibility to exchange their genetic material via the horizontal gene transfer process, i.e., exchanging pathogenic genes [17]. It was shown previously that the only groups of archaea involved in the dysbiosis of the human microbiota are the methanogenic archaea (named methanogens within the manuscript), which were found to be reduced in Crohn’s disease, ulcerative colitis, and severe acute malnutrition [18,19]. One of the most studied methanogens was *Methanobrevibacter smithii*, which is so far the only archaea implicated in microbial dysbiosis, including vaginosis and urinary tract infection [20]. Recently, a study performed on a Chinese population showed enrichment of halophilic archaea and depletion of methanogens in CRC compared to healthy subjects [13].

Among the archaea clades detected recently in the human gut microbiota [21] is the DPANN (Diapherotrites, Parvarchaeota, Aenigmarchaeota, Nanoarchaeota, and Nanohaloarchaeota) superphylum proposed in 2013 [22], characterized by its small cell size, limited metabolic capacities [23], and unique genetic characteristics [24]. The lack of genes associated with essential metabolic pathways in these microorganisms indicates a potential dependence on symbiotic relationships with other organisms, or even a parasitic mode of existence [25]. While their symbiotic or parasitic lifestyles could increase the chances for horizontal gene transfer with potentially pathogenic organisms, especially given their prevalence in tumor microenvironments [26], there is currently no established evidence linking them to colorectal carcinogenesis.

A prior study linked specific archaeal metabolites, such as short-chain fatty acids, indoles, and arsenic, to carcinogenesis. Metabolites such as H2 and trimethylamine (TMA) have been linked to carcinogens, specifically H2S and trimethylamine N-oxide (TMAO). These carcinogens are implicated in processes like inflammation, DNA damage, genotoxicity, and disrupted signaling pathways [21]. It is worth noting, however, that these metabolites can also be produced by other microorganisms, such as bacteria [27,28]. Thus, the unique functional signature of archaea in the gut microbiome of colorectal cancer remains to be discovered.

In our study, we delved into the relationship between the archaeome and colorectal cancer by examining 390 published fecal metagenomics datasets from both CRC patients and healthy individuals. We employed bivariate and multivariate analyses to explore the associations of the DPANN archaeon (*Candidatus Mancarchaeum acidiphilum*) with other detected archaeal species. Furthermore, we conducted a functional analysis to pinpoint the unique archaeal functional signatures present in the gut of CRC patients.

## 2. Materials and Methods

### 2.1. Published Fecal Metagenomic Data

Published metagenomes of 390 fecal specimens were examined, including samples from 192 CRC patients and 198 randomly selected controls (Table 1). Datasets of published fecal metagenomes are available from the European Nucleotide Archive (ENA) database “https://www.ebi.ac.uk/ena/browser/home” (accessed on 20–25 April 2022). Bioproject accession numbers are presented in Table 1. The clinical characteristics are shown in Appendix A.

### 2.2. Bioinformatic Processing and Biostatistical Analysis

#### 2.2.1. Taxonomic Analysis

##### Archaea Identification

The assignment of taxonomic labels to the published metagenomic data of 390 fecal samples was performed with Kraken, which is a taxonomic sequence classifier that labels short DNA reads with taxonomic labels [35]. This was performed by looking at the k-mers in a read and then querying a database with those k-mers. In a taxonomic tree of all genomes containing that k-mer, every k-mer in Kraken’s genomic library was mapped to its lowest common ancestor (LCA). After that, the set of LCA taxa corresponding to a read’s k-mers was analyzed to produce a single taxonomic label for the read, which could be any node in the taxonomic tree [35]. The taxonomic analysis of the metagenomic data was performed using the Galaxy Europe platform and its tool “Kraken” by mapping them against the archaea database “archaea 2020”; we used default parameters. The outputs of Kraken were then assembled into one single output “Kraken abundance table” (Appendix A), using the “Kraken taxonomic report” tool “https://usegalaxy.eu/” (accessed on 20 April to 2 May 2022).

##### Data Filtering and Archaeal Species Patterns in Colorectal Cancer

Data filtering was applied to remove low-quality features from the data, as very small numbers in very few samples were likely due to sequencing errors. To reduce false positives resulting from sequencing errors or low-level contamination, taxonomic characteristics were filtered based on their minimum count (set at four) and prevalence (at least 20% of samples containing four or more counts). Additionally, features that remained constant throughout the experiment were removed by filtering based on low variance (10% of features, determined using the inter-quantile range). Following filtering, library size differences were adjusted using cumulative sum scaling (CSS) normalization. Data filtering and normalization were performed using microbiomeAnalysit “https://www.microbiomeanalyst.ca/MicrobiomeAnalyst/home.xhtml” (accessed on 3–5 May 2022). Spearman correlation was used to calculate the correlation between archaeal species, CRC, and healthy phenotypes. A *p*-value less than 0.05 (typically ≤ 0.05) was statistically significant. Statistical analyses were performed using SPSS for Windows 20.0 (SPSS 20.0 for Windows; SPSS Inc., Chicago, IL, USA).

##### Correlation Analysis of DPANN Archaea and Other Archaeal Species

The Spearman’s correlation coefficient was calculated to estimate the possible associations between the DPANN archaeon *Candidatus Mancarchaeum acidiphilum* and the different archaeal species detected in the gut microbiomes of healthy subjects and CRC patients (Appendix A). A *p*-value less than 0.05 (typically ≤ 0.05) was statistically significant. Statistical analyses were performed using SPSS for Windows 20.0 (SPSS 20.0 for Windows; SPSS Inc., Chicago, IL, USA).

##### Multivariate Analysis

A multivariate analysis was performed to identify the association between *Candidatus Mancarchaeum acidiphilum,* considered an independent factor, and other archaeal species significantly associated with CRC or healthy phenotypes and found to be strongly correlated with the DPANN archaeon following bivariate analysis, which are *Methanocella conradii*, *Natrinema* sp. *J7-2*, *Ferroglobus placidus*, *Methanobrevibacter olleyae*, *Sulfolobus* sp. *E5-1-F*, *Pyrolobus fumarii*, and *Thermococcus chitonophagus* (Appendix A). Statistical analyses were performed using SPSS for Windows 20.0 (SPSS 20.0 for Windows; SPSS Inc., Chicago, IL, USA). A two-sided *p*-value of less than 0.05 was considered statistically significant.

#### 2.2.2. Data Assembly

De novo assembly of the metagenomic data was performed using the MEGAHIT tool in the Galaxy Europe platform https://usegalaxy.eu/ (accessed on 1–6 June 2023). MEGAHIT is a computational tool for ultra-fast assembly of large and complex metagenomics sequencing data [36]. The adjustment of assembly parameters was carried out following the instructions of the tutorial provided by the Galaxy Training Network [37]. Briefly, in the basic assembly options, the minimum k-mer size was adjusted to 21, the maximum k-mer size was adjusted to 91, and the increment of k-mer size for each iteration was fixed at 12.

#### 2.2.3. Functional Analysis

##### Genome Annotation

A functional study was then carried out using the contigs generated by the MEGAHIT tool. For this investigation, we utilized Prokka, a rapid prokaryotic genome annotation tool hosted on the Galaxy Europe platform. Prokka is used to predict and annotate coding sequences, rRNAs, tRNAs, and other genomic features. It annotates microbial genomes by integrating several bioinformatics tools, including Prodigal, RNAmmer, Aragorn, SignalP, and Infernal [38]. To map the contigs against only archaeal genes, we adjusted the “kingdom” option to “Archaea” “https://usegalaxy.eu/” (accessed on 5–7 June 2023).

##### Association of Archaeal Genes with Colorectal Cancer

The set of archaeal genes found in the studied metagenomic data was then analyzed using Spearman’s correlation test in Python [39] to establish their relationship with the CRC phenotype. The genes found to be correlated with CRC were then mapped against the UniprotKB database (https://www.uniprot.org/), accessed on 8–10 June 2023, to identify the genes that are mostly or only found in archaeal species. The identified genes were mapped against Uniprot-Gene Ontology (GO) database (https://www.uniprot.org/help/gene_ontology), accessed on 12–14 June 2023, to find their biological function.

## 3. Results

### 3.1. Study Characterization and Data Processing

We analyzed published fecal metagenomic data generated from metagenomic shotgun sequencing of fecal samples collected from 192 CRC patients and 198 healthy controls (Table 1, Appendix A). The pan-archaeome included 7 archaeal phyla, 11 archaeal classes, and 294 archaeal species found in both CRC and healthy samples (Appendix A). A total of 62 low-abundance features (archaeal species) were removed based on prevalence, and 25 low-variance features were removed based on an inter-quantile range (IQR); 216 features remained after data filtering (Appendix A). For the archaeal phyla and archaeal classes, the filtering step eliminated one and two low-variance features, respectively (Appendix A).

### 3.2. Archaeal Communities Associated with CRC Microbiomes

The taxonomic analysis revealed the presence of six phyla in the gut microbiomes of healthy and CRC subjects, which are Euryarchaeota, Thermoproteota, Nitrososphaerota, Candidatus Geothermarchaeota, Candidatus Micrarchaeota, and Candidatus Lokiarchaeota. A strong correlation between Candidatus Micrarchaeota, a phylum belonging to the superphylum DPANN, and CRC patients was highlighted (*p* < 0.001). Euryarchaeota was also correlated with the CRC phenotype, unlike Candidatus Geothermarchaeota, which was found to be associated with healthy subjects (Table 2).

In our study, we detected the presence of nine classes after the filtering step of the data presented by Archaeoglobi, Thermoplasmata, Methanococci, Halobacteria, Methanobacteria, Thermoprotei, Nitrososphaeria, Methanomicrobia, and Thermococci. We found a significant positive correlation between Archaeoglobi and CRC patients (*p* = 0.017) (Table 3).

The Spearman correlation showed a significant association between the different *Halobacteria* sp. and CRC phenotype. *Halapricum salinum*, *Halomicrobium* sp. *LC1Hm*, *Halosimplex rubrum*, *Salinibaculum litoreum*, *Haloferax gibbonsii*, *Halopenitus persicus*, *Halorubrum* sp. *RHB-C*, *Halorubrum trapanicum*, *Halostagnicola larsenii*, and *Natrinema* sp. *J7-2* were positively and highly correlated with CRC gut microbiomes (*p* < 0.001) (Appendix A). As for *Haloferax alexandrines* and *Halorubrum lacusprofundi*, these halophilic archaea were highly correlated with healthy subjects (*p* < 0.001). Other than halophilic archaea, *Methanoculleus bourgensis*, *Methanocella conradii*, *Methanosphaerula palustris*, and *Thermococcus* sp. were correlated with CRC patients (*p* < 0.001). *Methanobacteria* sp. and *Methanomicrobia* sp. were more abundant in healthy subjects’ gut microbiomes than in CRC patients. From the Methanobacteriaceae family, we highlighted a strong correlation (*p* < 0.001) between healthy subjects and *Methanothermobacter wolfeii*, *Methanobrevibacter olleyae*, *Methanobacterium paludism*, and *Methanobacterium* sp. *MZ-A1*. In contrast, *Methanobrevibacter smithii* showed a high correlation with CRC patients (*p* < 0.001). Regarding *Methanomicrobia* sp., correlation analysis revealed a strong and significant association (*p* < 0.001) between healthy gut microbiomes and *Methanosarcina barkeri*, *Methanosarcina flavescens*, and *Methanosarcina siciliae*. Two archaeal species belonging to the Archaeglobi class showed different correlations with the CRC phenotype; *Ferroglobus placidus* was found to be highly correlated with CRC patients (*p* < 0.001), and *Archaeoglobus veneficus* was correlated with healthy subjects (*p* = 0.025). Our analysis revealed the detection of a DPANN archaeon, presented by *Candidatus Mancarchaeum acidiphilum*, and its correlation with the CRC gut microbiome (*p* = 0.002) (Appendix A).

### 3.3. DPANN Archaeon Association with the Different Archaeal Species Detected in Healthy and CRC Subjects

Bivariate analysis showed a significant positive association (*p* < 0.001) between *Candidatus Mancarchaeum acidiphilum* and archaeal species positively correlated with CRC gut microbiomes (Section 3.2): *Methanocella conradii*, *Methanoculleus bourgensis*, *Natrinema* sp. *J7-2*, *F. placidus*, and *Methanohalophilus halophilus*. However, the DPANN archaeon was negatively and significantly correlated (*p* < 0.001) with archaea previously revealed to be associated with healthy subjects (Section 3.2): *Sulfolobus* sp. *E5-1-F*, *Pyrolobus fumarii*, *Thermococcus chitonophagus*, *Methanobrevibacter olleyae*, and *Methanobacterium* sp. *MZ-A1* (Appendix A). The multivariate analysis showed a highly significant correlation between *Candidatus Mancarchaeum acidiphilum* and both *Natrinema* sp. *J7-2* and *F. placidus* (*p* < 0.001) (Table 4).

### 3.4. Archaeal Proteins Identified in the Gut Microbiomes of Healthy Individuals and CRC Patients

Following annotation using Prokka, we identified 2427 archaeal proteins belonging to different functional categories, including enzymes (e.g., ribosomal RNA small subunit methyltransferase H, protein-arginine kinase, and glycine dehydrogenase subunits 1 and 2), structural proteins (e.g., DNA-binding protein and membrane-bound lytic murein transglycosylase F), and transport proteins (e.g., molybdate/tungstate-binding protein WtpA). Additionally, we noted the presence of regulatory proteins such as protein-arginine kinase and DNA mismatch repair protein MutL, as well as DNA/RNA-related proteins like DNA polymerase PolB subunit 25 and error-prone DNA polymerase. Proteins involved in metabolic processes, such as formate-tetrahydrofolate ligase and galactose-1-phosphate uridylyltransferase, were also detected, along with proteins involved in protein modification, including the NAD-dependent protein deacylase sirtuin-5, mitochondrial (Appendix A).

### 3.5. Association of Proteins Exclusively Detected in Archaea with Colorectal Cancer

To understand the potential implications of the detected archaeal proteins in CRC, we performed a Spearman correlation analysis. We found that 704 of the proteins exhibited either a positive or negative correlation with CRC (Appendix A). The genes showing a significant association with the CRC phenotype (Appendix A) and having relative abundance in the studied metagenomic data (Appendix A) were mapped against the UniProtKB database to identify genes predominantly or exclusively detected in archaea (Appendix A). We mapped the identified archaeal genes against GO annotations to discern their biological roles. Significantly, we observed a negative correlation between CRC and the archaeal protein, beta-galactosidase BgaH (*p* = 0.002), which plays a role in galactose metabolic process. Similarly, there was a negative association with the methanogen homoaconitase large subunit (*p* = 0.019), an archaeal protein involved in the biosynthesis of leucine. On the other hand, the polyferredoxin protein MvhB, associated with energy metabolism, showed a positive correlation with CRC (*p* = 0.045) (Table 5).

## 4. Discussion

CRC presents a public health problem; its incidence rate is estimated to increase from 1,931,590 new cases in 2020 to 3,154,674 by 2040 [1]. To prevent this huge increase, several scientific researchers have been trying to understand the mechanism of colorectal carcinogenesis, its initiation, and its development. Despite the multifactorial nature of colorectal cancer (CRC), the gut microbiota were found to be correlated with this pathology [40]. Unlike the bacteriome, which is the most analyzed component of the gut microbiota, it was only in 2020 that the fecal archaeal DNA of CRC patients was profiled and compared with that of healthy subjects from China [13].

Here, we employed several user-friendly methods to analyze the gut archaeomes of CRC patients and healthy individuals in a large cohort from different populations, including Galaxy, MicrobiomeAnalysit, and SPSS, for bioinformatics and statistical analyses. Galaxy offers a diverse set of bioinformatic tools specifically designed for metagenomic analysis that allow reproducible analysis workflows, collaboration, and complexity without programming experience [41]. Meanwhile, MicrobiomeAnalysit enables complex analysis without bioinformatic expertise or programming knowledge. Together, Galaxy and MicrobiomeAnalysit provide a comprehensive and in-depth analysis of metagenomic data, accommodating users of varying experience levels [42]. We used a Python script for the functional analysis to investigate any potential relationships between the discovered genes and CRC. This method was required due to the large number of identified genes which were too numerous for SPSS to analyze.

Taxonomically, after metagenomic data filtering using MicrobiomeAnalysit, a total of 217 archaeal species belonging to different phyla (Euryarchaeota, Thermoproteota, Nitrososphaerota, Candidatus Geothermarchaeota, Candidatus Micrarchaeota, and Candidatus Lokiarchaeota) were found in healthy and CRC patients. Interestingly, we detected the presence of Candidatus Micrarchaeota belonging to the DPANN superphylum, and we revealed its strong positive correlation with CRC gut microbiomes (*p* < 0.001).

Statistical tests performed to compare the abundance of archaeal species between CRC patients and healthy subjects showed the enrichment of methanogens in healthy gut microbiomes, except for *M. smithii,* which was found to be strongly correlated with the CRC phenotype. Previous studies identified the human gut archaeome and found that it consisted mostly of methane-producing archaea (methanogens), especially Methanobacteriales [43]. Regarding the abundance of *M. smithii* in the CRC gut microbiomes, previous research showed conflicting results, and it was reported that this archaeon was associated with Indian CRC patients [44], unlike the Chinese [13] and American [45] populations, which showed its association with the healthy phenotype. We also highlighted the abundance of *Halobacteria* sp. in the CRC gut microbiomes, confirming the previous study of Coker et al. (2020), in which they demonstrated that *Natrinema* sp. *J7-2* exhibited a gradual increase in abundance from the control group to patients with adenoma and then to those with CRC; furthermore, they proposed an association between the enrichment of some Haloarchaea species (*Halorubrum* sp., *Natrinema* sp., and *Halococcus* sp.) in CRC and the intake of salty food [13]. Several studies have reported a positive correlation between the consumption of salty foods and an increased risk of developing CRC [46,47,48]. This salty food habit, by promoting the proliferation of Haloarchaea, can disrupt the intestinal microbiome not only by altering the availability of nutrients on which intestinal bacteria depend for their growth and survival but also by producing “haloarcheocines”, which are antibiotics produced by the halophile archaea [49]. Another study in which a metagenomic analysis was performed on formalin-fixed paraffin-embedded tissues from Tunisian CRC patients showed a significant positive correlation between Halobacteria in tumor tissues compared to healthy mucosa [50].

Archaeoglobi, an archaeal class, was found to be correlated with CRC gut microbiomes according to our statistical analysis. *F. placidus* belonging to the Archaeoglobi was highly correlated with CRC microbiomes (*p* < 0.001). This archaeon is the first hyperthermophile known to grow anaerobically by oxidizing acetate and reducing ferric iron (Fe^3+^) to ferrous iron (Fe^2+)^ [51]. Acetate has been shown to inhibit CRC cell proliferation and induce apoptosis [52], so its deficiency could be considered a risk factor for CRC. The reduction of ferric iron (Fe^3+^) to ferrous iron (Fe^2+^) can generate reactive oxygen species (ROS) [53]. In fact, the role of ROS in carcinogenesis remains controversial, as ROS can have both pro-cancerous and anti-cancerous effects depending on their concentration. At moderate concentrations, ROS can contribute to the initiation, development, and progression of malignant neoplasms through the promotion of cell proliferation, invasion, and angiogenesis. However, at high concentrations, ROS can cause cancer cell apoptosis, leading to tumor suppression [54]. Otherwise, other species of archaea (*Ferroplasma* sp. [55]) and bacteria (*Thiobacillus ferrooxidans* or *Gallionella ferruginea* [56] *Acidithiobacillus ferrooxidans* [57]) could oxidize Fe^2+^, which can keep a limited concentration of the ferrous iron, thus inducing low concentrations of ROS. Regarding ferric iron and its relationship to CRC and the gut microbiome, a previous study found a significant correlation between microbial siderophores and advanced stages of CRC; taken together [58], we suggest the involvement of gut microbes, presented essentially by siderophores-producing bacteria and *F. placidus* in the iron deficiency in CRC patients. Due to the essential role of iron in supporting immune functions, a lack of iron may lead to an elevated risk of developing cancer by affecting the process of tumor formation and immune surveillance, as well as potentially modifying the immune environment within the tumor [59].

A DPANN archaeon, *Candidatus Mancarchaeum acidiphilum*, was detected within the studied human gut metagenomic data. Previously, this archaeon was identified in acidic ecosystems [60]. Interactions between this archaeon and protuberances from another archaeal order, Thermoplasmatales, suggested potential metabolite, energy, and macromolecule (including DNA) exchange [61]. Although we did not find a significant correlation with Thermoplasmatales, we observed its positive association with *F. placidus* and *Natrinema* sp. *J7-2* in the CRC phenotype context. Evidence from other studies, including findings on *Thermotoga maritima* [62] and *Methylobacterium extorquens* [16,63], indicates that archaeal DNA can be shared among both archaeal and non-archaeal species. When considering the association of DPANN archaea with pathogens, there are risks related to the spread of genes contributing to carcinogenic properties, virulence, resistance, and metabolism [64]. Such transfers could influence gut microorganisms, potentially exacerbating CRC development. The prevalence of archaeal microorganisms within gut microbiota is limited. Still, their significance should not be overlooked. To illustrate this point, consider viruses: although they constitute only about 0.2% of gut microbiota, they can incorporate their genes into the human genome, potentially triggering oncogenes [65,66]. Remarkably, a significant portion of genes in archaea, especially in the DPANN group, are classified as “hypothetical” proteins or genomic “dark matter”, accounting for 30% to possibly 80% of their total content [67]. While these genes are still uncharacterized, it should be noted that they may be able to encode pathogenic phenotypes, underscoring the necessity of thorough functional genomics investigations. Characteristics of DPANN archaea, such as reduced genome sizes [22], may indicate fewer obstacles to gene transfer or integration. There is a conceivable, though distant, route for gene transfer if DPANN archaea interact with viruses also infecting eukaryotic cells [68]. Such viruses could assimilate genes from DPANN archaea and later infuse them into a eukaryotic host.

At a functional level, we aimed to understand the unique pathogenic potential of archaea by focusing on genes specific to them. This emphasis arises because many genes can be found across various cellular organisms, including bacteria, eukaryotes, and archaea. Hence, simply mapping metagenomic data against the functional database of archaea might not delineate the distinct role of archaea in colorectal carcinogenesis. When performing Spearman correlation then mapping against UniProtKB, we found a significant correlation with three proteins exclusive to archaeal species. Notably, one of these is the MvhB-type polyferredoxin. Although all organisms include a variety of ferredoxin proteins, the “MvhB-type polyferredoxin” variant is only found in methanogenic archaea and not in other archaeal groups or bacteria [69]. This protein acts as an electron transfer chain in potential redox–enzyme complexes and plays a pivotal role in electron transfer processes. It is composed of three to seven 2[4Fe-4S]-ferredoxin units [69]. Our findings highlight a significant positive association between this protein and CRC. Given that polyferredoxins have an iron–sulfur composition, their production is closely tied to iron availability. Thus, an increase in MvhB-type polyferredoxin production might signal enhanced iron uptake by archaea. Consequently, we propose a potential risk associated with increased archaeal iron intake in CRC, both taxonomically and functionally. We also revealed that BgaH was inversely correlated with CRC. Delving into BgaH’s enzyme function, we found it primarily targets β-d-galactosides lactulose as key substrates [70], which suppresses harmful bacteria and enhances the production of beneficial metabolites like short-chain fatty acids (SCFAs) [71]. We also identified a negative correlation between the methanogen homoaconitase large subunit and the prevalence of CRC. This enzyme is critical for the biosynthesis of leucine, an amino acid recently found to be inversely associated with CRC as well [72]. Our findings are consistent with a prior study in which a decrease in microbial genes associated with the production of branched-chain amino acids, specifically leucine, valine, and isoleucine, was highlighted during carcinogenesis [73].

Our study presents the first metagenomic analysis of the gut microbiome in Tunisian CRC patients. We focused on sequencing DNA taken from tumor-adjacent FFPE tissues from these patients. Significantly, the discovery of archaeal sequences in our samples underscores the viability of conducting an archaeome analysis from FFPE tissues using an affordable DNA extraction protocol and easy-to-use bioinformatic tools. Moreover, our findings highlight a notable enrichment of Halobacteria in the tumor mucosa when compared to the normal mucosa in these CRC patients. Based on these findings, it is important to highlight that the gut microbiome has been shown to have the ability to influence treatment response in several ways, including drug metabolism, immunomodulation, inflammation, and the tumor microenvironment [74,75]. Our findings might open the door for the creation of therapeutic and preventative strategies for colorectal cancer given this wider context.

In terms of limitations, our study’s inability to conduct a comprehensive comparison of archaeomes across the various stages of colorectal carcinogenesis has limited our insight into their evolutionary patterns and influence on disease progression. Additionally, a significant number of genes in archaea are classified as “hypothetical” proteins or termed genomic “dark matter.” As the exact functional roles of these genes remain undetermined, further research is urgently needed to discern their impact on human health and disease.

## 5. Conclusions

In conclusion, we explored the role of gut archaea in colorectal carcinogenesis and identified significant associations between CRC and specific archaeal species, notably *Ferroglobus placidus* and *Natrinema* sp. *J7-2*. Additionally, we detected the presence of the DPANN superphylum in the gut, indicating potential gene transfers that might impact CRC progression. We further emphasized the significance of certain proteins unique to archaeal species, such as the polyferredoxin protein MvhB, Beta-galactosidase BgaH, and Methanogen homoaconitase large subunit, underscoring the importance of comprehending the distinct pathogenic potential of archaea in CRC.

## Figures and Tables

**Table 1 microorganisms-11-02742-t001:** Summary of study populations.

Project n°	CRC	Healthy	Country	Reference
PRJEB7774	20	24	Austria	[29]
PRJEB27928	42	39	Germany	[30]
PRJNA447983	22	22	Italy	[31]
PRJDB4176	33	35	Japan	[32]
PRJEB10878	46	48	China	[33]
PRJNA397112	29	30	India	[34]

**Table 2 microorganisms-11-02742-t002:** Correlation analysis of archaeal phyla and CRC phenotype.

Phylum	Spearman’s Correlation Coefficient (*r*)	*p*-Value
Candidatus Micrarchaeota	0.199	0.000
Candidatus Geothermarchaeota	−0.147	0.004
Euryarchaeota	0.118	0.020
Thermoproteota	−0.046	0.369
Candidatus Lokiarchaeota	0.005	0.916
Nitrososphaerota	−0.004	0.933

**Table 3 microorganisms-11-02742-t003:** Correlation analysis of archaeal classes and CRC phenotype.

Class	Spearman’s Correlation Coefficient (*r*)	*p*-Value
Archaeoglobi	0.1199	0.017844
Thermoplasmata	0.088356	0.081387
Methanococci	−0.08369	0.09889
Halobacteria	−0.05394	0.28799
Methanobacteria	0.042504	0.40256
Thermoprotei	−0.03954	0.43616
Nitrososphaeria	0.03813	0.45273
Methanomicrobia	−0.02483	0.62497
Thermococci	−0.02424	0.63325

**Table 4 microorganisms-11-02742-t004:** Multivariate analysis of DPANN archaeon and its association with other archaeal species detected in the gut microbiome.

Coefficients ^a^						
Model	Unstandardized Coefficients		Standardized Coefficients	Sig.	95.0% Confidence Interval for B	
	B	Std. Error	Beta		Lower Bound	Upper Bound
*Thermococcus chitonophagus*	0.186	0.405	0.032	0.647	−0.610	0.981
*Ferroglobus placidus*	1.868	0.288	0.278	0.000	1.301	2.435
*Natrinema* sp. *J7-2*	0.872	0.081	0.461	0.000	0.713	1.031
*Methanocella conradii*	0.076	0.109	0.030	0.487	−0.138	0.289
*Sulfolobus* sp. *E5-1-F*	−0.199	0.298	−0.355	0.505	−0.784	0.387
*Pyrolobus fumarii*	0.269	0.522	0.273	0.607	−0.758	1.296
*Methanobrevibacter olleyae*	0.073	0.095	0.081	0.444	−0.114	0.259

^a^ Dependent variable: Candidatus Mancarchaeum acidiphilum.

**Table 5 microorganisms-11-02742-t005:** Association of proteins mostly or exclusively detected in archaea with CRC.

Proteins	Spearman’s Correlation Coefficient (*r*)	*p*-Value	% of UniprotKB Search Results with Taxonomy: Archaea and Query: Protein	GO Annotation
Beta-galactosidase BgaH	−0.155543828	0.002	100%	Galactose Metabolic Process
Methanogen homoaconitase large subunit	−0.11878052	0.019	100%	Leucine biosynthetic process
Polyferredoxin protein MvhB	0.101673216	0.045	96%	Energy Metabolism

## Data Availability

All the data used for this study (metagenomic data) are already presented in the Appendix A and available online.

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
