# Peer review of "The Archaeome’s Role in Colorectal Cancer: Unveiling the DPANN Group and Investigating Archaeal Functional Signatures"

_microorganisms, 2023, doi:10.3390/microorganisms11112742_

Round 1
Reviewer 2 Report
Colorectal Cancer ranks fourth in the world among all newly diagnosed malignant neoplasms, after breast, prostate and lung cancer. The likelihood of developing cancer after age 50 doubles over each subsequent decade and peaks at age 75. There is a sharp increase in morbidity and mortality from colorectal cancer in age groups up to 34 years and 45 - 49 years. According to numerous data, in 60–70% of patients with newly diagnosed colorectal cancer, the tumor is detected in stages III-IV. As a result of late diagnosis, long-term results of colorectal cancer treatment remain today at the level of the 70s of the last century. The 5-year survival rate after surgical treatment of colorectal cancer remains at 50 - 55%. However, the rapid progress in knowledge about the molecular and biological nature of colorectal cancer has brought a lot of new information into the understanding of the pathogenesis of the disease and inspires hope for the development of effective primary prevention measures, improving diagnostic and treatment results.
This research was done and presented at a very high scientific level. The authors have done a great job. All work steps are described in detail.
I recommend in "Abstract: Background & Aims:" write the countries from which the patient samples were taken.
Please note: According to the International Code of Nomenclature, all generic and species names are written in italics. Please make all species and generic names in italics throughout the text.
